# Composite Flours Based on Black Lentil Seeds and Sprouts with Nutritional, Phytochemical and Rheological Impact on Bakery/Pastry Products

**DOI:** 10.3390/foods14020319

**Published:** 2025-01-18

**Authors:** Christine (Neagu) Dragomir, Sylvestre Dossa, Călin Jianu, Ileana Cocan, Isidora Radulov, Adina Berbecea, Florina Radu, Ersilia Alexa

**Affiliations:** 1Faculty of Food Engineering, University of Life Sciences “King Mihai I” from Timisoara, Aradului Street No. 119, 300645 Timisoara, Romania; christine.neagu@usvt.ro (C.D.); calinjianu@usvt.ro (C.J.); ileanacocan@usvt.ro (I.C.); florinaradu@usvt.ro (F.R.); ersiliaalexa@usvt.ro (E.A.); 2“Food Science” Research Center, University of Life Sciences “King Mihai I” from Timisoara, Aradului Street No. 119, 300645 Timisoara, Romania; 3Faculty of Agriculture, University of Life Sciences “King Mihai I” from Timisoara, Aradului Street No. 119, 300645 Timisoara, Romania; isidoraradulov@usvt.ro (I.R.); adina_berbecea@usvt.ro (A.B.)

**Keywords:** nutritional, phytochemical, MIXOLAB, lentil, sprout

## Abstract

This paper aimed to study the nutritional, phytochemical and rheological properties of some composite flours based on wheat flour (WF) mixed with non-germinated (LF) and sprouted lentil flour (SLF), in order to fortify the wheat flour and to obtain functional bakery/pastry products. The composite flours based on wheat flour and bean lentil flour (BLWF) and sprouted lentil flour (SLWF) were analyzed from the point of view of proximate composition (proteins, lipids, total carbohydrates, and minerals), content of individual and total polyphenols (TPC), as well as the contents of macro and microelements. For use in baking/pastries, the composite flours were tested from the point of view of rheological behavior using the MIXOLAB system, and the profiles obtained were compared with those of bread and biscuit. The results indicated that fortifying wheat flour with lentil flour, both in non-germinated and sprouted forms, increased the protein by 0.6–35.2% and mineral content of the samples and decreased the lipids by 8.3–43.2% and the carbohydrates by 2.8–9.4%. The total polyphenol content (TPC) increased by fortifying the wheat flour with non-germinated and sprouted lentil flour, the increase being between 39.2–131.4%. Regarding individual polyphenols, nine polyphenols were determined, of which epicatechin (46.979 mg/kg) and quercetin (45.95 mg/kg) were identified in the highest concentration in the composite flours. The increase in micronutrient intake by fortifying wheat flour with black lentil flour in both germinated and ungerminated form is more significant compared to the increases recorded in the case of the main macronutrients (Ca, Na, Mg, and K). The micronutrients increased in the composite flours in the order: Cu < Zn < Fe < Mn. The MIXOLAB profile highlighted that black lentil flour, although having a higher absorption index than that recommended for biscuit production, would improve the stability of the dough.

## 1. Introduction

Legumes are high-quality foods, able to replace proteins of animal origin, being a complete source of essential nutrients for people suffering from nutritional deficiencies [1].

Lentils (*Lens culinaris*) are one of the oldest cultivated plants, having a history of thousands of years as a staple food for various civilizations. Native to the Fertile Crescent region, which includes the Near East, lentils played an important role in human nutrition and the development of early agriculture [1].

In Europe, lentils have been cultivated for thousands of years due to their nutritional value and adaptability to various climatic conditions. Currently, the main lentil producing countries include India, Canada, Turkey and Australia. According to the Food and Agriculture Organization (FAO), world lentil production was approximately 6.3 million tons in 2020, where total lentil production in Europe represented a small fraction of global production [1]. However, in recent years, there has been an increasing trend of interest in growing lentils in Europe, due to the increased demand for vegetable protein and sustainable agricultural practices [2].

European countries with a tradition of growing lentils include: France, Spain, Italy, and Greece. Although it is not a major producer at the European level, in Romania, lentils are cultivated in limited areas, especially in small households and on organic farms, where lentils can bring additional income to farmers, especially in the context of the increased demand for plant-based and organic foods [2].

Lentils have gained popularity in the global diet due to increased interest in healthy and sustainable diets. In addition, being a legume, lentils fix nitrogen in the soil, contributing to its fertility and reducing the need for chemical fertilizers. It can also be grown in various climates and soils, making it suitable for different European regions.

Due to its functional properties, lentil contributes to the prevention and management of chronic diseases such as diabetes and cardiovascular diseases, reduces LDL cholesterol and the intake of soluble fibers, reduces blood pressure due to the content of potassium and magnesium [3], manages the body weight and supports intestinal motility [4]. Lentils contain polyphenols such as flavonoids and tannins, which have antioxidant properties and protect cells against oxidative stress [5].

For healthy food preparation, lentils can be used in various forms (whole grains, flour, sprouts). Studies have shown that the consumption of lentils by age groups has a general trend toward youth and young adults, as they are more open to trying new foods and can include lentils in their diets, especially in the context of the popularity of vegetarian and vegan diets [1].

Recently, germination has been analyzed as an innovative approach in green food engineering, having the potential to increase the nutrient supply of plant matrices with promising uses in the fields of functional foods [6]. After germination, the sprouts become richer in nutrients and easier to digest compared to the seeds from which they come [1]. Germination transforms seeds into shoots, or seedlings, a process that causes significant changes in their nutritional and biochemical composition. Sprouted seeds are becoming very popular among health-conscious consumers worldwide who are looking for optimal nutrition [7]. Sprouting makes significant changes to the nutritional profile of lentils compared to dried grains. Lentil sprouts contain between 20–25% high-quality protein, with a full spectrum of essential amino acids [7].

Sprouting can increase protein digestibility by up to 30% while the fiber content remains significant, supporting digestive health [8]. The fat content of sprouts is low, but it slightly increases the concentration of essential fatty acids during germination. The concentration of B complex vitamins (B2, B3, B6 and folate) increase during germination [9], while vitamin C, absent in dry grains, is produced during germination. The content of iron, zinc, magnesium and phosphorus in lentil sprouts becomes more bioavailable due to the reduction of phytic acid, an inhibitor of mineral absorption [9], and the production of polyphenols and flavonoids, which have antioxidant properties, is stimulated through germination [5]. Considering these facts, the introduction of sprouted lentils in functional nutrition represents a healthy alternative to conventional nutrition.

Bakery products, made mainly from wheat flour, are an important source of complex carbohydrates, but contain low amounts of dietary fiber and certain essential minerals. For these reasons [10], the fortification of bakery products by using in the manufacturing recipe composite flours based on wheat flour mixed with legumes in raw or sprouted form represents a viable alternative for obtaining functional foods.

Previous research on the possibility of fortifying wheat flour for baking with leguminous flour [11] emphasized the potential of increasing the nutritional intake by adding lupin flour, germinated [12] or non-germinated [12,13]; soybean [14]; pulse [15]; or chickpea [16]. Also, the characteristics of some composite flours based on wheat–legume flours, as well as the hypoglycemic effects of composite flours based on sprouted legumes [17] and their use in baking have been studied in other papers [10,18]. This study presents elements of novelty compared to the existing studies so far because it presents from an innovative perspective, with an applicative character, which recommends the optimal solution of lentil-based composite flour in germinated or non-germinated form, for the bakery and/or pastry industry, based on the cumulative results obtained regarding their nutritional, functional and rheological properties.

The purpose of this study was to obtain and characterize, from a nutritional point of view and in term of phytonutrients intake, composite flours based on wheat flour mixed in different proportions with lentil flour in raw or germinated form. In order to use these composite flours in baking and pastries, the rheological behavior of the dough was analyzed using the MIXOLAB system and the profile was compared with that of the ideal dough for bread and biscuits in order to formulate technological recommendations for the food industry. Also, correlations were drawn among the main compounds analyzed.

## 2. Materials and Methods

### 2.1. Sample Preparation and Seed Germination

Black lentils (*Lens culinaris*) were sourced from S.C. Pronat România, while wheat flour type 650 was purchased from Auchan Timișoara, Romania and produced by S.C.Boromir, Romania. This flour was selected for its common use in baking and its well-documented rheological properties. The lentil seeds were thoroughly washed and soaked in water for 12 h to facilitate and accelerate germination. After being placed in a tray near a window with natural light, germination occurred over approximately 2–3 days at a temperature range of 20–28 °C [19].

Germination began to occur, and the sprouts were harvested after 6–7 days. Their length was measured using a ruler, with harvesting carried out once they reached 8–10 cm. After cutting, the sprouts were placed in an oven (FroiLabo AC 60, Air Concept, Froilabo, Paris, France) at 50 °C for 5 days to ensure complete drying. To produce black lentil sprout flour, the dried sprouts underwent a controlled grinding process using a kitchen blender (Bosh TSM6013B, BSH Hausgerate, Germany), ensuring a uniform particle size and preserving nutrient integrity until a fine powder was obtained [20].

For the second blend, black lentil seeds were ground into a fine powder. The seeds used were the same ones utilized for producing the black lentil sprouts. Figure 1 presents the technological flow of the lentil seeds germination process.

### 2.2. The Preparation of Composite Flours

The composite flours were obtained by mixing wheat flour (WF) with black lentil seeds (LF) and wheat flour (WF) with black lentil sprouts (SLF) in different percentages in order to obtain six different mixtures. The flour blends, WF + LF (wheat flour + black lentil seeds) and WF + SLF (wheat flour + black lentil sprouts), were prepared by adding LF and SLF to WF in different proportions of 10%, 15%, and 20%, resulting in six samples as presented in Figure 2.

### 2.3. Determination of Proximate Composition of Composite Flours

The proximate composition of the samples was determined using standardized analytical ISO methods: humidity SR 91/2007 pct.10, protein SR EN ISO 8968-1:2014, lipids SR 91:2007 pct.14,4, and mineral substances SR ISO 2171/2010 [21].

Moisture content was measured by drying the samples at 105 °C until a constant weight was achieved, with results expressed as a percentage of the total sample weight.

Ash content was quantified by incinerating the samples in a muffle furnace (Naberthen GmbH, Lilienthai/Bremen, Germania) at 650°C until all organic matter was combusted, leaving only inorganic mineral residues.

Total fat content was determined using Soxhlet equipment (Velp Scientifica SER 148 Solvent Extractor, Italia) and petroleum ether as the solvent.

Protein content was calculated based on the nitrogen content obtained via the Kjeldahl method, using a conversion factor of 6.25. These analyses provided a detailed insight into the proximate composition of the samples, contributing to the comprehensive evaluation of their nutritional and functional properties.

Each sample was analyzed in triplicate and the results are presented as mean value ± standard deviation (sd).

The total carbohydrate content (%) was calculated using the following formula:carbohydrate (%) = 100 − (moisture + ash + fat + protein) (1)

This formula estimates the carbohydrate content as the remaining fraction of the sample after accounting for moisture, ash, fat, and protein.

### 2.4. Phytochemical Profile of Composite Flours

#### 2.4.1. Determination of Total Polyphenols Content (TPC)

The total polyphenol content (TPC) was quantified using the Folin–Ciocalteu method, as described in [12]. This method relies on the reduction of the Folin–Ciocalteu reagent by polyphenols in the sample, resulting in the formation of a blue complex that can be measured spectrophotometrically [22].

For the analysis, first the alcoholic extract of the sample was prepared by adding 10 mL ethanol (70%)(Merck, Darmstadt, Germany) to 1 g sample and stirred rigorously for 30 min on a shaker (IDL, Freising, Germany). Then, 0.5 mL of the extract was mixed with the Folin–Ciocalteu reagent (Sigma–Aldrich Chemie GmbH, Munich, Germany), diluted 1:10 (*v*/*v*) with distilled water and incubated for 5 min at room temperature, followed by the addition of 1 mL sodium carbonate 60 g/L (Geyer GmbH, Renningen, Germany). The reaction mixture was then allowed to develop color by incubation at 50 °C for 30 min in a thermostat (INB500, Memmert GmbH, Schwabach, Germany). Absorbance was measured at 750 nm using a UV–Vis spectrophotometer Specord 205 (Analytik Jena AG, Jena, Germania).

The TPC was calculated using a calibration curve with a standard of gallic acid (Merck, Darmstadt, Germany) in the range of concentration between 2.50–200 µg/mL and the results were expressed in milligrams of gallic acid equivalents (GAE) per 100 g of sample. All measurements were conducted in triplicate. The coefficient of correlation for the standard curve obtained was 0.9986, reflecting the method’s reliability [23].

#### 2.4.2. Determination of Individual Polyphenols by HPLC

A Shimadzu Chromatograph equipped with SPD-10A UV and LC-MS 2010 detectors, EC 150/2 NUCLEODUR C18 Gravity SB 150 × 2 mm × 5 μm column was used. The chromatographic conditions were as follows: mobile phase A: water acidified with formic acid at pH 3, B: acetonitrile acidified with formic acid at pH 3, gradient program: 0.01–20 min 5% B, 20.01–50 min 5–40% B, 5–55 min 40–95% B, 55–60 min 95% B. The solvent flow rate was 0.2 mL/min at 30 °C. The monitoring wavelengths were 280 nm and 320 nm. The calibration curves were performed in the range of 20–50 μg/mL. The results are expressed in mg/kg.

### 2.5. Determination of Macro and Microelements Using Atomic Absorption Spectroscopy (AAS)

Macro and microelements were determined by SAA after calcination of the sample in a calciner (Naberthen GmbH, Lilienthai/Bremen, Germany) until the ash reached a white color. The ash was dissolved in 20% HCl, filtered and used for the determination of macro and microelements by the AAS method [23]. For the calibration curve, a mixed standard solution (ICP Multi Element Standard solution IV CertiPUR) was used. The minimum detection limits (MDL) for the analyzed elements were 0.02 ppm for Mg and K; 0.06 ppm for Fe, Cu, Zn, and Mn; and 0.03 ppm for Ca. The results are expressed in ppm. All determinations were made in triplicate [16].

### 2.6. Rheological Determination of Composite Flours Using MIXOLAB System

Rheological comparisons of the flour properties of different mixtures of BLWF and SLWF were studied with the help of the “Chopin+” device of the MIXOLAB equipment (Chopin, Paris, France) in accordance with the ICC Standard Method [12] and the “Chopin+” protocol [24].

For analysis, 50 g of flour sample were placed into the MIXOLAB bowl for mixing. After blending the solid ingredients, the equipment automatically added the required amount of water to achieve optimal consistency. The MIXOLAB test parameters were set as follows: a mixing speed of 80 rpm, with the first plateau lasting 8 min at a temperature of 30 °C. The second plateau had a duration of 7 min at 90 °C, and the third plateau lasted 5 min at 50 °C. The temperature gradients for the first and second phases were both set at 4 °C/min.

The MIXOLAB profile provided the following evaluations: water absorption, dough development time, dough stability (mixing resistance), and maximum torque during mixing (C1). It also measured protein weakening (C2) caused by mechanical stress and increasing temperature, the rate of starch gelatinization (C3), minimum torque during heating (C4), and torque after cooling to 50 °C (C5) [12].

### 2.7. Statistical Analysis

All determinations were made in triplicate and the results are reported as mean values ± standard deviation (SD). Differences between means were analyzed with one-way ANOVA, followed by multiple comparison analysis using the *t*-test (two-sample assuming equal variances) using Microsoft Excel 365. Differences were considered significant when *p*-values < 0.05. The PCA statistical analysis (Principal Component Analysis) was conducted using the Past3 program.

## 3. Results and Discussion

### 3.1. The Proximate Composition of Composite Flours

The results regarding the proximate composition of the composite flours are presented in Table 1.

Analysis of the results from Table 1 reveals that WF had a higher moisture content than the samples with different proportions of black lentil flour, and was in the range of samples with sprouted black lentil flour. WF had a moisture content corresponding to 11.43 ± 0.39%, whereas the moisture content of samples containing black lentil/wheat flour and sprouted black lentil/wheat flour, respectively, ranged from 10.77 ± 0.06% to 9.92 ± 0.13% and from 11.39 ± 0.01% to 11.86 ± 0.02%. These results are similar to those of [25], who reported that the moisture content of breads obtained with different proportions of black lentils was lower than that of the control bread [25].

In terms of mineral content, the gradual addition of black lentils and lentil sprouts significantly improved the mineral content of the various flours compared with WF. The mineral content rose from 0.5 ± 0.02% in WF to 2.62 ± 0.03% in BLWF3 and 5.51 ± 0.16% in SLWF3. Similar results were obtained by Hernandez-Aguilar et al., 2020 [26]. In their study, the mineral content rose from 0.89 to 0.96% between the control sample and the sample with 10% sprouted lentils. Çelik and İlyasoǧlu (2022) revealed an improvement in mineral content between the control sample and that with 20% black lentils [25]. It can therefore be concluded that the addition of black lentils to wheat flour improves the mineral composition.

It was also found in the present study that flours containing sprouted lentils were richer in minerals than those containing black lentil flour. The protein content followed the same pattern as the mineral content. In fact, as the quantity of black lentils, whether sprouted or not, increased in the composition of the compound flours, so did the protein content. Compound flour samples with black lentils had 12.86 ± 0.02, 13.98 ± 0.09 and 15.06 ± 0.13% protein for BLWF1, BLWF2 and BLWF3, respectively. Samples containing different quantities of sprouted lentils had 11.7 ± 0.11, 12.34 ± 0.2, and 13.31 ± 0.17% protein for SLWF1, SLWF2, and SLWF3, respectively. This suggests that the progressive addition of black lentil flour or sprouted black lentil flour to wheat flour would significantly improve the protein content. Similar results were obtained by other authors [25,26]. It should also be noted that flour composed of different proportions of black lentils was richer in protein than flour containing the latter’s sprouts.

Protein supplementation of flour products by adding legumes is an important practice for improving the nutritional value of these products. Flour products, such as bread, biscuits or pasta, are usually high in carbohydrates but low in protein and essential amino acids. The addition of legume flour (chickpeas, lentils, beans, or peas) completes the amino acid profile, especially with lysine, which is missing from cereals [27,28]. Protein supplementation promotes a more diversified and sustainable diet, with a positive impact on food security.

In terms of lipids and carbohydrates, the wheat flour provided 1.30 ± 0.02% lipids and 75.63 ± 0.39 g/100 g carbohydrates. The gradual addition of black lentil flour or sprouted black lentil flour to wheat flour would reduce its levels of lipids and carbohydrates. Similar results have been reported by other authors [12,22,25,26].

Figure 3 shows the increases/decreases of nutritional parameters in composite flours (BLWF and SLWF) compared to type 650 wheat flour intended for a bakery. There is an important increase in the protein content that varies between 15.4–35.2% depending on the percentage of wheat flour substituted with lentil flour (BLWF), between 0.6–14.5% in the case of fortification with 10–20% of sprouted lentil flour (SLWF). The lipid content, however, decreases with the substituted fraction, the decrease being significant and between 25.4–43.2% depending on the percentage of black lentil flour added, and between 8.3–10.6% when replacing wheat flour with lentil sprout flour.

The carbohydrate intake provided by the consumption of 100 g of composite flour decreased, but in a smaller proportion (2.8–5.3%) compared to the lipid fraction, in the fortified flours with black lentils, with a decrease between 2.5–9.4% of the carbohydrate content for the SLWF samples compared to the intake generated by white flour.

The importance of the use of legumes in flour formulas and their role in controlling metabolic functions and obesity has been highlighted in other studies [29]. Legumes help reduce the risk of chronic diseases such as type 2 diabetes, cardiovascular disease and obesity due to their high fiber and phytonutrient content [30]. Fiber and protein from the legume fraction slow down the absorption of carbohydrates, which helps keep blood sugar stable.

### 3.2. The Phytochemical Profile of Composite Flours

In the Figure 4 are presented the values regarding the TPC content of the composite flours and Table 2 presents the profile of individual polyphenols identified in the BLWF and SLWF mixtures.

The total polyphenol content of the composite flours fortified with sprouted and non-sprouted black lentil flour ranged from 583.723 mg/100 g in the case of the BLWF1 sample to 791.237 mg/100 g in the case of the SLWF3 sample. Through germination, the content of total polyphenols increased, the increase being between 89.22–122.8%. Other authors have reported a content of 379.76 mg/100 g [31] and 456 mg/100 g [19] in non-germinated lentil seeds, between 461.28–788.88 mg/100 g [31] and between 486–614 mg/100 g [19] in lentil sprouts at different stages of germination. The variety of lentils has a significant influence on the content of total polyphenols. A recent study carried out on different varieties of lentils from Romania showed that yellow lentils are distinguished by a lower content of total polyphenols (69.00 mg/100 g), followed by red lentils (85.89 mg/100 g), brown lentils (247.56 mg/100 g), and black lentils (248.38 mg/100 g), and the highest value was recorded in the case of green lentils (327.31 mg/100 g) [32].

Much higher values (1200 mg/100 g) for total polyphenol content have been reported in other studies [10], which emphasizes that sprouting lentils can lead to a significant increase in TPC, with values rising by up to 185% over a 7-day germination period.

In the analyzed composite flours, hydroxycinnamic acids such as rosmarinic, coumaric, ferulic and beta-resorcylic acid were identified, as well as polyphenols from the flavonoid, flavonol and stilbene categories (Table 2).

Caffeic and gallic acids were not quantified above the minimum detection limit of the method for any of the analyzed samples. In wheat flour (WF), the most abundant hydroxycinnamic acid was beta-resorcylic acid (3.77 mg/kg), followed by rosmarinic acid (1.386 mg/kg), while feluic and coumaric acids were not found in the sample. In composite flours based on wheat flour with the addition of sprouted lentil flour, an increase in the content of hydroxycinnamic acids can be observed with the amount of wheat flour substituted with sprouted lentil flour. The maximum value for hydroxycinnamic acids in the samples of the flour composition with the addition of sprouted lentils was recorded in the sample with 20% sprouted lentil flour, SLW3, with values of 35.280 mg/kg for beta-resorcylic acid, 4.789 mg/kg for coumaric acid, 22.65 mg/kg for ferulic acid and 30.059 mg/kg for rosmarinic acid.

In the composite flour samples based on a mixture of wheat flour and ungerminated lentil flour (BLWF), the content of hydroxycinnamic acids was lower and varied for resorcilic acid from 3.880 mg/kg (BLWF1) to 4.037 mg/kg (BLWF3), from 3.489 mg/kg (BLWF1) to 4.012 mg/kg (BLWF3) for acid coumaric, from 2.316 mg/kg (BLWF1) to 3.981 mg/kg (BLWF3) for ferulic acid, and from 2.614 mg/kg (BLWF1) to 8.490 mg/kg (BLWF3) for rosmarinic acid.

Rosmarinic acid is less common in lentils, but is found in very small amounts, especially in the sprouts. In a 6-day-old sprout, the rosmarinic acid content can reach approximately 28–43 mg/kg, depending on germination conditions [6]. Ref. [31] highlights the fact that by germination, the synthesis of rosmarinic acid in the lentil sprouts can be induced.

The way germination takes place also influences the accumulation of hydroxycinnamic acids in the sprouts. It was highlighted that germination in the dark for 6 days induces the highest increase in ferulic acid, reaching a concentration of 82.47 mg/kg DW [6]. Another study highlighted the increase of ferulic acid from 6.45 mg/kg to 10.32 mg/kg after a 6 day sprouting period [31].

β-Resorcylic acid (2,4-dihydroxybenzoic acid) is a less common phenolic compound but with notable biological properties, including antioxidant activity, anti-inflammatory and antifungal potential. The content of β-resorcylic acid in dry lentil seeds is usually below 1 mg/100 g dry weight, being a minor compound in the phenolic profile [5]. Germination causes a modest increase in β-resorcylic acid concentration due to the activation of enzymes that break down cellular components and release bound phenols. The reported values are about 0.5–1.5 mg/100 g dry weight, depending on the duration of germination (3–7 days) [1].

The caffeic acid content of lentils is about 0.5–2 mg/100 g dry weight [5] and increases to 5.0 mg/100 g dry weight, depending on the duration of germination [33]. In our study, caffeic acid was undetectable in agreement with the data reported by other authors [7,31].

The coumaric acid content of lentils is about 0.3–1.5 mg/100 g dry weight, and lentil germination can lead to a significant increase in coumaric acid levels, with values of about 1.0–3.0 mg/100 g dry weight [34]. The content of coumaric acid in brown and black ungerminated lentil seeds *Lentils* (*Vicia lens* (L.) *Coss. & Germ.*), from Greece varied between 12–22 mg/kg and increased from 278 mg/kg to 372 mg/kg after germination, when the sprout length reached 15 cm [19].

In raw lentils, resveratrol is reported only in trace or very small amounts, below 0.1 mg/100 g dry weight, depending on the variety and growing conditions [7], while in sprouted lentils the resveratrol content ranges from 0.05–0.2 mg/100 g dry weight, depending on the germination time (3–7 days) and environmental conditions [1]. Higher concentrations were determined in sprouted lentils (19.00–80.64 mg/kg) depending on the germination conditions [31].

In our study, the content of resveratrol in flours fortified with germinated and ungerminated lentils was higher in the wheat flour sample and varied between 7.274–34.086 mg/kg for the samples with the addition of ungerminated lentils (BLWF) and between 28.576–64.693 mg/kg for the composite flours with the addition of germinated lentils (SLWF) in agreement with the data reported by Barakat et al., 2022 [31].

Quercetin is a flavonoid with strong antioxidant, anti-inflammatory and anti-carcinogenic properties, being present in various foods, including legumes. Germinating lentils for 3–5 days leads to an optimal concentration of quercetin, while prolonged germination (>7 days) may result in decreased concentrations. Quercetin is present in raw lentils in amounts of about 0.5–1.5 mg/100 g dry weight and increases to 1.5–3.5 mg/100 g after germination, depending on the environmental conditions (temperature, humidity and light exposure) [1]. Higher concentrations were reported in the lentil sprouts (35.29–54.12 mg/kg) after 6 days of germination at 20 °C and a relative humidity of 90–93% [31].

In our study, a higher concentration of quercetin was detected in the raw lentil sprouts (74.919 mg/kg). Wheat flour is poor in quercetin, and its fortification with lentil flour, whether as such or germinated, leads to a significant increase in the level of this bioactive compound. The addition of 20% sprouted lentils leads to a quercetin level of 45.956 mg/kg, while in the version fortified with unsprouted lentils in a proportion of 20%, we found 12.869 mg/kg quercetin.

### 3.3. The Macro and Microelements Composition of Composite Flours

The macro and microelements composition of flours obtained with seed lentil flours and sprouted lentil flours is presented in Table 3, and the increase is expressed as the percentage of the content of the macro and microelements by fortifying the wheat flour with lentil flour in the two forms as in Figure 5.

The fortification of wheat flour for baking with black lentil flour, either in ungerminated or germinated form, led to an increase in the contents of macro and micronutrients of the composite flours, which increased with the fraction of substituted wheat flour. With the exception of calcium, for which lower values were recorded for the SLWF1 and two variants fortified with sprouted flour compared to the BLWF1 and two variants based on non-sprouted lentil flour, for all other elements there was an increase in mineral content through the addition of lentil flour in the two forms analyzed. The recorded values were between 302.367–324.777 mg/kg for the varieties of flour fortified with lentil seeds and between 281.884 and 305.635 mg.kg for the composite flour samples based on lentil germinated flour. The increase in calcium content in the composite flour was maximal in the case of the addition of 20% lentil flour from seeds (128.57%) and 115.10% in the case of fortification with sprouted lentil flour (SLWF3).

Sodium is the only macroelement that was found in greater quantity in wheat flour (190.096 mg/kg) compared to composite lentil flour (69.131–102.208 mg/kg) and composite flours with 10–15% lentil sprouts (86.655–121.798 mg/kg) (Figure 5).

Magnesium recorded values between 419.525 mg/kg for the BLWF1 sample and 623.333 mg/kg, the maximum value recorded for the SLWF3 sample. The increases in Mg content in composite flours by the addition of germinated flour compared to wheat flour for baking were up to 55.36%, and up to 38.15% in the case of substituting wheat flour with 30% black lentil flour, which indicates a possibility of covering the dietary deficit in this macroelement by using composite flours based on germinated or ungerminated lentil flour.

Potassium did not register significant increases in the composite flours in relation to wheat flour (5.96–10.89%). The maximum value recorded was 559.637 mg/kg in the sample fortified with 20% sprouted lentil flour, while in unfortified wheat flour, the K content was 504.647 mg/kg.

The analysis of the magnesium content of lentil seeds from different varieties has been reported in other studies, identifying values between 850–1060 mg/kg depending on the variety. Lower values, between 671–950 mg/kg, were recorded for potassium, and between 480–1280 mg/kg for calcium [7]. Other authors reported a level of 900–1000 mg/kg Mg and 700–1000 mg/kg for Ca [18], while for sprouted lentils a level of 140–400 mg/kg K, 70–90 mg/600 mg/kg Ca and 820 mg/kg Mg [33].

Our results are in agreement with those reported by other authors regarding the influence of germination on the content of macroelements in lentils [10], highlighting the fact that the fortification of flour samples with germinated lentils has a greater positive impact on comparative Ca with the other analyzed macroelements (Mg and K). Santos et al., 2020 reported that the magnesium concentration was not affected by sprouting, while Ca and K had percentage increases between 41% and 58%, and 28% and 30%, respectively, in the best performing varieties [10].

Other studies highlighted that the addition of germinated lupin flour in a proportion of 10–30% in wheat flour for biscuits led to a 50.35% increase in the content of K and 59.64% for Ca [12], while for composite flours, with the addition of 30% lupin flour, the contents were 615.80 mg/kg Mg, 287.66 mg/kg Ca and 110.67 mg/kg K, which led to a substantial increase in the content of macroelements in the bread obtained based on lupin composite flour [12]. Lentils are especially rich in essential microelements such as iron, manganese and zinc. Fortifying flour with lentil flour can increase the content of these minerals in baked goods, helping to prevent dietary deficiencies of these minerals.

The increase in micronutrient intake by fortifying wheat flour with black lentil flour in both germinated and ungerminated form was more significant compared to the increases recorded in the case of the main macronutrients (Ca, Na, Mg, K). The micronutrient increases in the composite flours increased in the order: Cu < Zn < Fe < Mn.

Manganese is the microelement that registered the most significant increase, varying between 573.00% in the case of substitution with 10% lentil flour (BLWF1) and 843.64% when 20% lentil flour was added to wheat flour. The recorded value for Mn in the case of type 650 wheat flour was 2.088 mg/kg and it reached a maximum value of 19.700 mg/kg in the BLWF3 variant. Ref. [10] reported values between 11–25 mg/kg Mn in different varieties of lentils in germinated or ungerminated form [10].

The second microelement that recorded significant increases through the fortification of wheat flour was Fe, for which increases were recorded between 133.52% for the BLWF1 sample and 375.66% for the SLWF3 sample. These increases in the level of Fe through the partial substitution of wheat flour with germinated or non-germinated lentil flour suggests the possibility of using lentil flour in obtaining functional and dietary foods intended for people with iron deficiency anemia. Previous results reported regarding the Fe content in different varieties of sprouted lentils and lentil seeds indicated an intake of 34–66 mg/kg Fe, with slight increases recorded in the case of sprouted samples [35]. Martinez et al., 2023, approached the study of the fortification of wheat flour with Fe by adding lentil flour, emphasizing the fact that a substitution of 25% lentil flour for wheat flour led to an increase in the Fe content from 11.97 mg/kg to 26.42 mg Fe [10].

Flour fortified with sprouted or seeded black lentils increased the Zn content of the samples, the increase being 34.60% in the case of the LWF1 sample and up to 221.90% for the SLWF3 sample. The effect of germination on the Zn content of different lentil varieties indicated values between 36–51 mg/kg in germinated seeds and values between 31–39 mg/kg for non-germinated grains [18].

### 3.4. Rheological Properties of Composite Flours

The MIXOLAB CHOPIN Technologies was used to determine the rheological characteristics of the different types of flour used in this study. The MIXOLAB CHOPIN is a dough characterization instrument designed to evaluate flour’s quality and baking performance. It provides a comprehensive view of the dough’s behavior during the baking process and measures several key parameters such as water absorption capacity, mixing stability, initial heat resistance, viscosity variation during heating, viscosity at high temperature, and viscosity during cooling. This instrument simulates real conditions of flour used in bakery products, analyzing dough consistency and protein and starch quality [24].

In the present study, wheat flour was used as a control flour compared to different composite flours of wheat/black lentil and wheat/black lentil sprouts. The curves obtained for each of the flours analyzed are presented in Figure 6, Figure 7 and Figure 8.

The water absorption of flour is a key parameter that must be strictly observed to obtain good-quality bakery products. Figure 9 shows the results obtained for water absorption for the different types of composite flours analyzed.

Analysis of the results showed that water absorption was lower for the control sample (WF) than for the other samples with different proportions of black lentil flour and black lentil sprouts. For samples containing black lentil flour, water absorption varied between 59.9% and 60.1%. There was also an increase in the water absorption value with increasing quantities of black lentils in the samples. The same observation was made for composite flours containing various levels of substitution of wheat flour by black lentil sprouts. The water absorption values were 56.9%, 57%, and 58.6%, respectively, for SLWF1, SLWF2, and SLWF3. Similar results were obtained by Dabija et al., 2017 [36] when studied the effect of yellow pea flour addition on wheat flour dough and bread quality. In this study, the water absorption capacity of composite wheat/yellow pea flours varied from 57.2 to 58.5 for substitution rates ranging from 5 to 20%. The addition of legumes changes the structure and properties of the dough, influencing the final characteristics of bakery products [36]. Legume flours have a higher water absorption capacity due to their high protein and fiber content, resulting a firmer and less stretchy dough. Adding legumes can prolong the time it takes to get a homogeneous dough, as the proteins in legumes take longer to hydrate [37].

Figure 10 shows the stability time obtained for each of the doughs produced from different flours.

Analysis of the results from Figure 10 showed that WF obtained the best stability time (11.45 min). This value is higher than those obtained by [12,22,31,37], which varied between 8.02 and 9.52. The stability of the dough decreased for composite flours, both those based on lentil flour (BLWF) and those based on sprouted lentils (SLWF) compared to the stability of the dough obtained only from wheat flour type 650. The decrease in the stability of the dough in flours composed with legumes has been highlighted in other studies [12,36]. However, in the case of the addition of sprouted lentils, the decrease was less pronounced compared to the BLWF samples, with a slight increase in the stability of the dough through the addition of 30% sprouted lentils (10.52 min), which emphasizes the possibility of processing composite flours based on sprouted lentils in bakeries.

The values of different torques are presented in the Figure 11 and represent the maximum torque during mixing (C1), protein weakening (C2) caused by mechanical stress and increasing temperature, the rate of starch gelatinization (C3), minimum torque during heating (C4), and torque after cooling to 50 °C (C5).

C1 is a characteristic for the first part of the MIXOLAB profile that defines the mixing parameters at 30 °C; for this parameter, there were no significant differences among the values obtained for WF, BLWF2, BLWF3, and SLWF3.

The second stage is the first heating stage and corresponds to the weakening of the protein network of the dough. The combined effect of mechanical shear stresses and the temperature constraint induces a decrease in the C2 torque, which represents the minimum value of the torque produced by the passage of the dough subjected to mechanical and thermal constraints due to the beginning of protein destabilization [12]. Compared to WF, the BLWF and SLWF composite flours were characterized by a lower C2 torque, values that were significantly lower in the case of SLWF (C2: 0.332–0.366 Nm) compared to BLWF (C2: 0.545–0.471 Nm). Similar results regarding the behavior of composite flours based on a wheat–triticale flour mixture have been reported in other studies [38].

In the third stage of the MIXOLAB profile, starch gelatinization takes place. Heating the dough induces the swelling of the starch granules and increases the viscosity of the dough. The boiling phase describes the behavior of the starch and is characterized by a maximum torque produced during the heating phase (C3) [39]. This parameter also registers significant decreases due to the addition of a legume matrix, especially in the case of SLWF samples (C3: 1.302–1.426 Nm) compared with WF (C3:1.917 Nm).

C4 is calculated as the ratio of torque after a holding time at 90 °C and the maximum torque during the heating period [24] and reflects the enzymatic activity of the dough. A high content of alpha-amylase induces an increase in the value of C4 and β [40]. The C4 value decreases in composite flours in this order: WF > BLWF > SLWF, which reflects a low alpha-amylase activity, leading to a lower cleavage of starch into fermentable sugars during the dough fermentation process.

In the stage 4, the dough is subjected to the double action of mixing and enzymatic activity. A decrease in consistency may be observed at this stage. Cooling of the dough (step 5) leads to the degradation of the starch and the stiffness of the dough increases. C5 is the difference between the torque produced after cooling to 50 °C and that after the heating period and was at its maximum for WF (C5 = 3.490 Nm) and minimum for SWFL (C5 = 2.007 Nm).

Figure 12, Figure 13 and Figure 14 show the indices of WF’s MIXOLAB profile for samples of flours blended with black lentil and black lentil sprouts, compared with the indices recommended by MIXOLAB for bread production. Analysis of these figures showed that the absorption index recommended by MIXOLAB for bread production was between eight and nine. Of all the flours analyzed, only those with different proportions of black lentils (BLWF) had an absorption index close to that recommended by MIXOLAB. There was no change from one sample to the next. They all obtained an index equivalent to seven. On the other hand, the composite flours with different proportions of black lentil sprouts (SLWF) obtained an index equivalent to three. Given that the absorption index is the MIXOLAB parameter used to measure a flour’s water absorption capacity [24,39], the composite flours with different proportions of black lentils would have a water absorption capacity very close to that required for bread production, compared with flours composed of different proportions of black lentil sprouts.

According to the MIXOLAB handbook [24], the mixing index is a parameter that measures the stability of the dough and, in particular, the development time and weakening of the dough during kneading at 30 °C. Analysis of the results obtained in the present study showed that flours with different proportions of lentil flours obtained mixing index values within the range recommended by MIXOLAB for bread production. The same observation was made for the gluten index and for viscosity where only the SLWF3 sample reached the optimal value. On the other hand, for samples with different proportions of black lentil sprouts (SLWF), only the 20% sample (SLWF3) obtained the recommended value for the mixing index. Also, the amylase index was in the range of optimal values for all SLWF samples. For the rest of the indices studied, none of the samples reached a recommended value, including the control sample. Hernandez-Aguilar et al., 2020 [26], in their study regarding the use of lentil sprouts as a nutraceutical alternative for the elaboration of bread, highlighted that the bread obtained with the addition of increasing amounts of black lentil sprouts was harder and less cohesive than the control bread. In addition, the quality of the dough and flour was modified by the addition of sprouted lentils and this effect increased with an increasing concentration of sprouted lentils [24]. Similar results were also obtained by Turfani et al., 2017 [41], who studied the technological properties of wheat bread enriched with lentil or carob flours. That study concluded that lentil flour reduces the toughness, stability and resistance of wheat dough [26].

Considering the results obtained by comparing the MIXOLAB profile of the analyzed samples with that of the optimal rheological profile for bread making, we can say that the BLWF3 sample with 20% lentil flour, which overlaps four of the three analyzed indices, is optimal from the point of view of the rheological behavior of the dough used in bakeries.

Figure 15, Figure 16 and Figure 17 show the indices recommended by MIXOLAB for the production of biscuits for each sample. WF would be a good flour for the production of biscuits, since it would have obtained very good values for each of the indices studied, especially for the absorption, gluten, amylase and retrogradation indices. As for the samples with different proportions of lentil flour, all the samples obtained a good mixing index for biscuit production up to 15%; the samples had good amylase and retrogradation indices. Black lentil flour, although having a higher absorption index than that recommended for biscuit production, would improve the stability of the dough and the development time and weakening of the dough in composite flours. On the other hand, black lentil sprouted flour overlapped with the optimal model for the retrograde index for all experimental variants and for the mixing parameter for the SLWF3 variant.

### 3.5. Correlation Between Analyzed Compounds

Figure 18 and Figure 19 present the Pearson correlation matrix between individual polyphenol contents and macro and microelement contents for composite flours BLWF (Figure 18) and SLWF (Figure 19).

The analysis of the Pearson correlations between the studied variables: individual polyphenols and macro and microelements related to wheat flour-based composite flours mixed with lentil flour in a proportion of 10–20% (BLWF experimental variants) indicated that there was a strongly positive correlation between: TPC and epicatechin (r = 0.94), TPC and quercetin (r = 0.96), TPC and ferulic acid (r = 0.96), TPC and rosmarinic acid (r = 0.93) and TPC and coumaric acid (r = 0.96), which demonstrated that these polyphenols contribute significantly to the total polyphenols determined. Conversely, there was a strong negative correlation (r = −0.33) between TPC and rosmarinic acid.

Regarding the correlations between the analyzed macroelements and the individual polyphenols, it was observed that Ca has strong positive correlations with variables such as quercetin (r = 0.92), coumaric acid (r = 1:00) and resveratrol (r = 1.00), which indicates a synergistic relationship between calcium and these compounds.

Mg had a profile similar to Ca and showed strong positive correlations with the same phenolic compounds. K showed significant positive correlations with epicatechin (r = 1:00), ferulic acid (r = 0.96), rosemary acid (r = 0.98) and quercetin (r = 0.93), indicating an essential role in samples with high antioxidant activity.

The microelements analyzed (Zn, Fe, Mn, and Cu) generally presented strongly positive correlations with the majority of individual polyphenols, except for beta-resorcilic acid, which presented negative or weakly positive correlations with all microelements.

In the case of SLWF composite flours there were also positive correlations between the analyzed components, but these were weaker compared to those obtained in the case of BLWF variants. Thus, between macroelements and polyphenolic compounds there were strongly positive correlations between the pairs: K/coumaric acid (r = 0.99), K/quercetin (r = 0.97), K/TPC (r = 0.98), and K/ferulic acid (r = 0.85). Mg was strongly positively correlated with TPC and ferulic acid (r = 1), resveratrol (r = 0.97), and quercetin (r = 0.98), and Ca was strongly positively correlated with coumaric (r = 1), TPC and quercetin (r = 0.98).

Among the microelements, Fe, Zn, Mn, Cu were strongly positively correlated (r > 0.85) with TPC, coumaric acid, quercetin and resveratrol, while Cu and Ni showed strongly positive correlations with epicatechin (r > 0.9).

Na was the macroelement that showed negative correlations with most of the other parameters analyzed, which indicates that samples with a high sodium content have lower levels of polyphenols and essential macroelements and microelements.

In addition, the multi-parametric statistical evaluation method using PCA (Figure 20 and Figure 21), were used for further analysis of the data regarding individual polyphenols and macro and microelements. Based on a linear correlation matrix, PCA was applied to the mean values of the measured traits to study which parameters contributed the most to the total data variation. For the individual polyphenols evaluation, Principal Component 1 (PC1) accounted for 87.383% of the total variance, while Principal Component 2 (PC2) accounted for 8.674% of the variance. It was observed that variables such as resveratrol and quercetin were the most influential in the differentiation of samples, having long vectors oriented toward positive PC1, with the caffeic, coumaric, beta-resorcilic and ferulic variables having a lower impact on differentiation with short vectors.

From Figure 20 it can be observed that the WF sample is positioned far on the negative side of PC1, which indicates a low content of polyphenols. BLWF1, BLWF2, and BLWF3 are grouped near the origin, in the central area of the graph, indicating a balanced content of polyphenols such as coumaric, beta-resorcilic, and ferulic. SLWF1 and SLWF2 are positioned on the negative side of PC2 and they show differences from BLWF, but they remain closer in terms of chemical composition, while SLWF3 is distinct, indicating a high content of major polyphenols. Vectors that form a small angle to each other (e.g., resveratrol and quercetin) indicate a strong positive correlation between these variables, while variables with opposite or perpendicular vectors have a low or negative correlation.

The PCA analysis of the projection of the macro and microelement parameters of the composite flours (Figure 21) indicates that PC1 explains 65.593% of the variance and PC2 explains 30.449% of the variance, reflecting the secondary variation.

From the interpretation of the biplot, it can be seen that WF is located in the negative part of Component 1 and Component 2, suggesting a low content of the variables related to these vectors (K, Mg, and Fe). Samples BLWF1, BLWF2, and BLWF3 are grouped in the negative part of PC2, indicating chemical similarities, their content being more balanced in trace elements such as Zn and Fe. SLWF3 is clearly positioned in the positive part of PC1 and PC2, indicating a high content of Mg, K, and other elements. SLWF2 also lies on the positive side of PC1, suggesting a similar but less extreme profile than SLWF3. SLWF1 is positioned between BLWF and SLWF3, indicating an intermediate composition. The vectors of the Mg, K, and Ca variables are long, which suggests a significant influence on the variation explained by Component 1, while Na is positioned almost perpendicular to Component 2, which indicates a more modest influence on PC2. Zn and Fe contribute to the differentiation of the samples on Component 2, having vectors oriented in the direction of the BLWF samples.

## 4. Conclusions

The global evaluation from the nutritional, phytochemical and rheological point of view of the composite flour fortified with lentil flour (BLWF) and sprouts (SLWF) showed that these formulas present a valid vegetal source of nutrients with application in the bakery/pastry industry.

The nutritional and phytochemical profile of the composite flours was improved compared to wheat flour, being distinguished by a higher percentage of proteins, phenolic compounds, macro and microelements and by a lower lipid fraction and carbohydrate content. Sprouted lentil composite formulas (SLWF) are richer in phytonutrients compared to lentil flour composite flours (BLWF), but from the point of view of rheological behavior in bakery/pastry manufacturing technology, the dough behavior is more deficient. Considering the results obtained by comparing the MIXOLAB profile of the analyzed samples with that of the optimal rheological profile for bread making, we can say that the sample with 20% lentil flour is optimal from the point of view of the rheological behavior of the dough used in the bakery and with less content (10–15%) for obtaining biscuits.

Our study strengthens the idea of widening the assortment range of vegetal matrices for use in baking in order to find alternative solutions to cereal flour.

## Figures and Tables

**Figure 1 foods-14-00319-f001:**
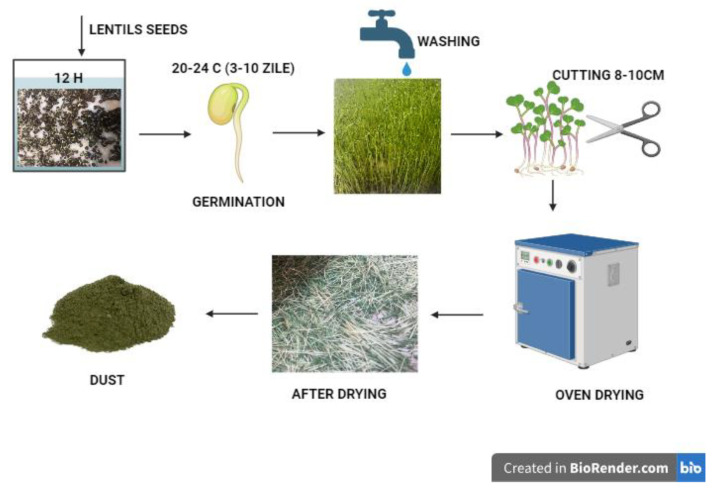
Technological flow for obtaining black lentil sprouts. Figure created with BioRender.com, accessed on 5 December 2024.

**Figure 2 foods-14-00319-f002:**
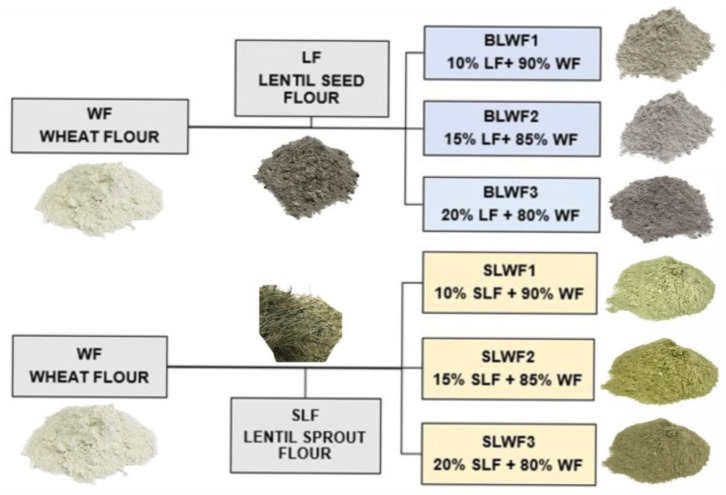
The composite flours: BLWF composite flours obtained by mixture of WF + LF; SLWF composite flours obtained by mixture of WF + SLF. Figure created with BioRender.com, accessed on 5 December 2024.

**Figure 3 foods-14-00319-f003:**
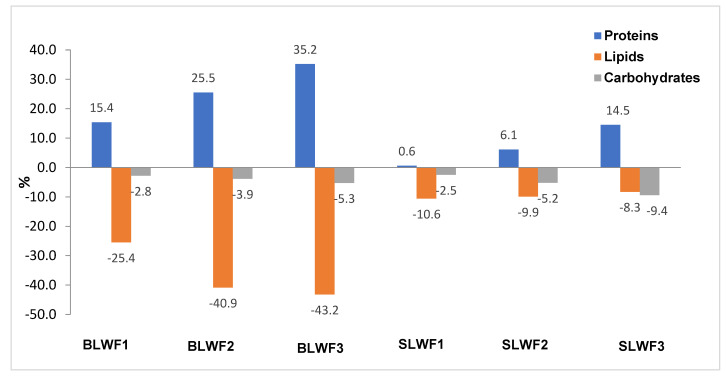
The increase/decrease of nutritional parmaeters in composite flours compared with wheat flour type 650. WF-wheat flour, BLWF1–3-composite wheat–lentil flours, SLWF1–3-composite wheat–lentil sprouts flours.

**Figure 4 foods-14-00319-f004:**
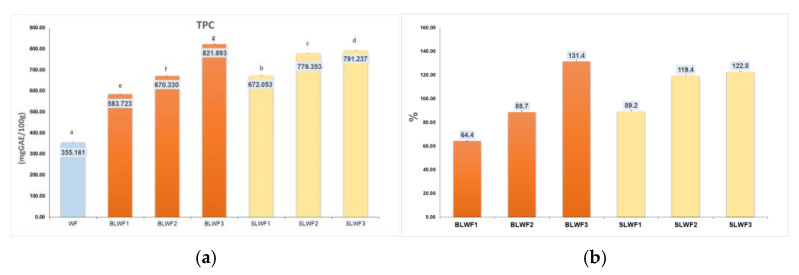
(**a**) TPC (mg GAE/100 g) of composite flours; (**b**) the increase in TPC content (%) of composite flours compared with wheat flour type 650. WF-wheat flour, BLWF1–3-composite wheat–lentil flours, SLWF1–3-composite wheat–lentil sprouts flours. The values are expressed as mean values ± standard deviations of all measurements; data within columns sharing different superscripts are significantly different (*p* < 0.05); data within the columns sharing the same superscripts are not significantly different (*p* > 0.05).

**Figure 5 foods-14-00319-f005:**
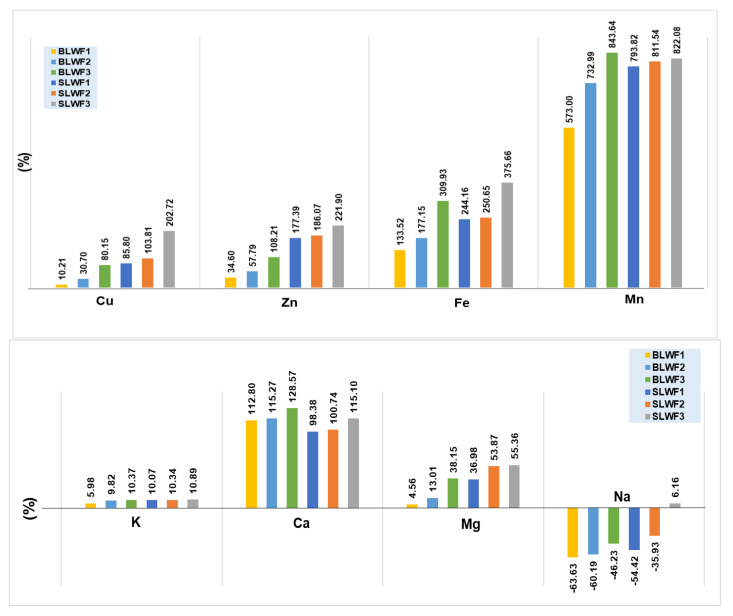
The increase/decrease of macro and microelements in composite flours compared with wheat flour type 650. WF-wheat flour, BLWF1–3-composite wheat–lentil flours, SLWF1–3-composite wheat–lentil sprouts flours.

**Figure 6 foods-14-00319-f006:**
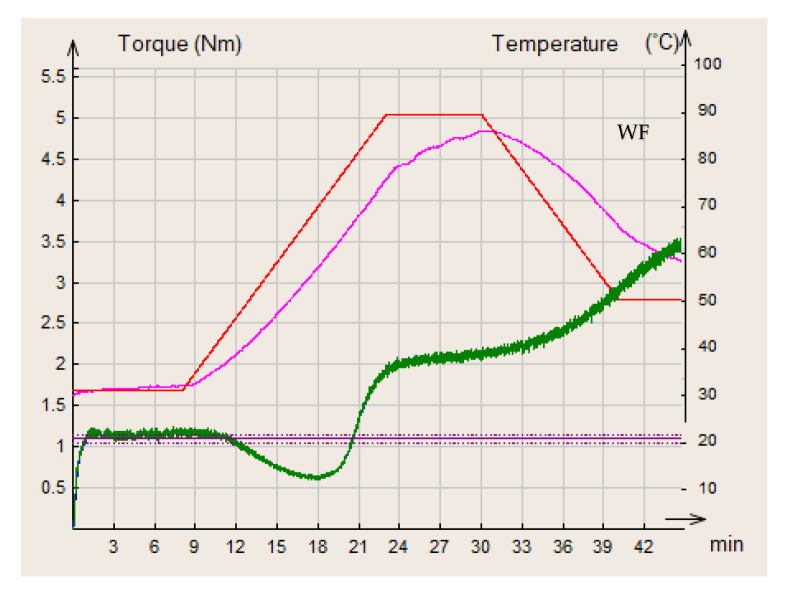
MIXOLAB rheological profiles of the analyzed sample with 100% wheat flour (WF). Red line—MIXOLAB temperature (°C), pink line—dough temperature (°C), green line—MIXOLAB curve.

**Figure 7 foods-14-00319-f007:**
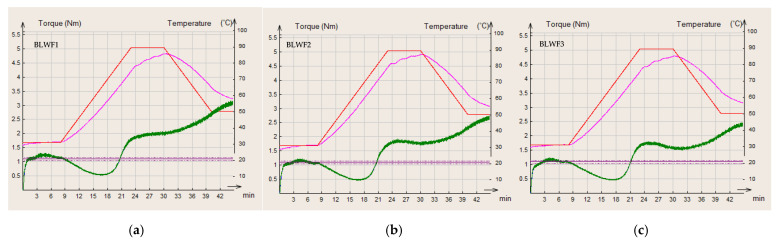
MIXOLAB rheological profiles of the composite flours with different proportions of black lentils flour and wheat flour type 650: (**a**) BLWF 1, (**b**) BLWF2, (**c**) BLWF3. red line—MIXOLAB temperature (°C), pink line—dough temperature (°C), green line—MIXOLAB curve.

**Figure 8 foods-14-00319-f008:**
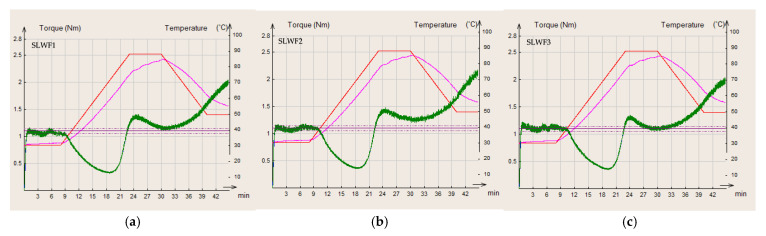
MIXOLAB rheological profiles of the composite flours with different proportions of black lentil sprouts flour and wheat flour type 650: (**a**) SLWF 1, (**b**) SLWF2, (**c**) SLWF3. red line—MIXOLAB temperature (°C), pink line—dough temperature (°C), green line—MIXOLAB curve.

**Figure 9 foods-14-00319-f009:**
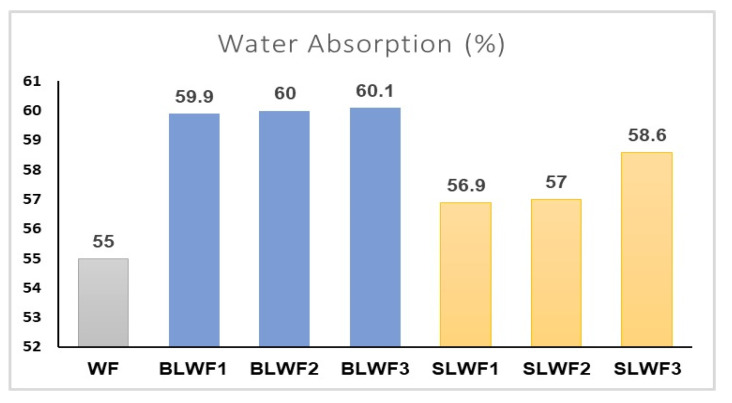
Water absorption (%) of composite flours determined using MIXOLAB system. WF-wheat flour, BLWF1–3-composite wheat–lentil flours, SLWF1–3-composite wheat–lentil sprouts flours.

**Figure 10 foods-14-00319-f010:**
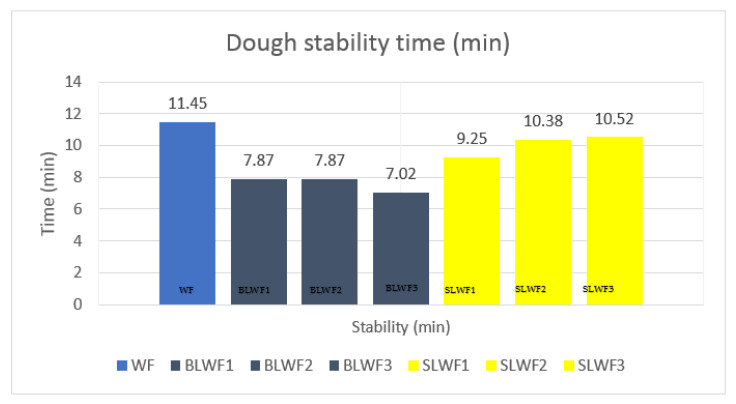
Dough stability time (minutes) of composite flours determined using MIXOLAB system. WF-wheat flour, BLWF1–3-composite wheat–lentil flours, SLWF1–3-composite wheat–lentil sprouts flours.

**Figure 11 foods-14-00319-f011:**
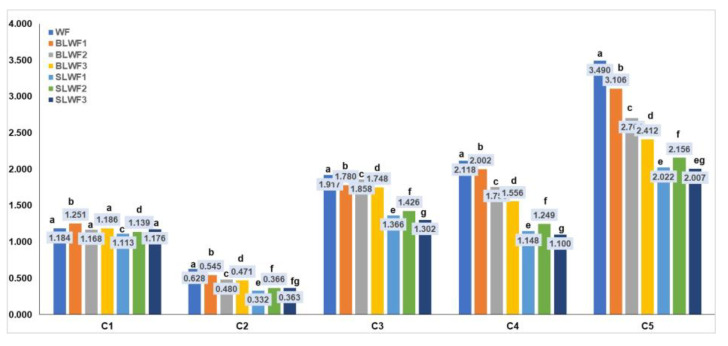
Torque indices (Nm) for composite flours (BLWF, SLWF) and wheat flour type 650 (WF). C1: maximum torque during mixing; C2: torque reflecting protein weakening caused by mechanical stress and increasing temperature; C3: torque reflecting rate of starch gelatinization; C4: minimum torque during heating; C5: torque after cooling to 50 °C. WF-wheat flour, BLWF1–3-composite wheat–lentil flours, SLWF1–3-composite wheat–lentil sprouts flours. The values are expressed as mean values ± standard deviations of all measurements; data within the each group columns sharing different superscripts are significantly different (*p* < 0.05); data within the each group columns sharing the same superscripts are not significantly different (*p* > 0.05). * nd—not detectable.

**Figure 12 foods-14-00319-f012:**
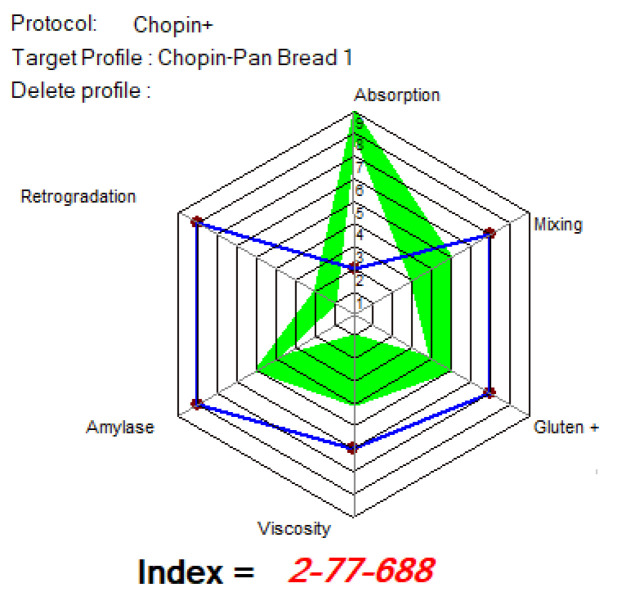
MIXOLAB Profiler index of the analyzed sample with 100% wheat flour (WF) for bread technology.

**Figure 13 foods-14-00319-f013:**
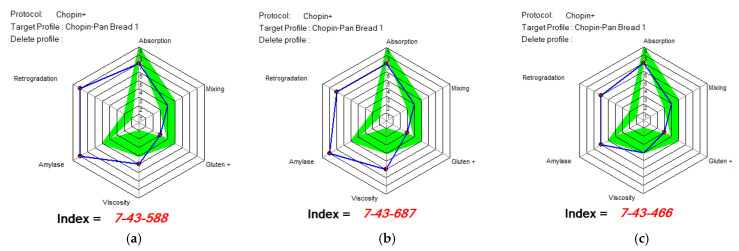
MIXOLAB Profiler index of composite flours with different proportions of black lentil flour (BLWF) and wheat flour type 650 (WF) for bread technology. (**a**) BLWF 1, (**b**) BLWF2, (**c**) BLWF3. Blue line represents the profile of composite flours and green line represents the profile of optimal MIXOLAB parameters for bread technology.

**Figure 14 foods-14-00319-f014:**
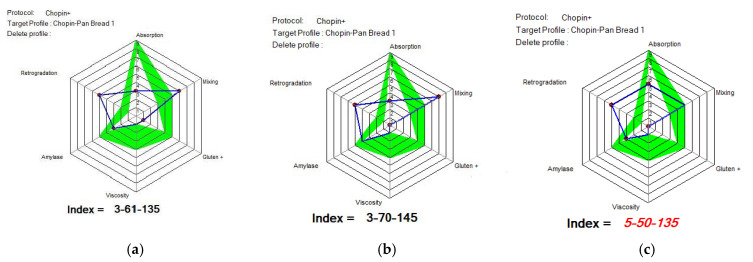
MIXOLAB Profiler index of the composite flours with different proportions of black lentil sprouts flour (SLWF) and wheat flour type 650 (WF) for bread technology. (**a**) SLWF 1, (**b**) SLWF2, (**c**) SLWF3. Blue line represents the profile of composite flours and green line represents the profile of optimal MIXOLAB parameters for bread technology.

**Figure 15 foods-14-00319-f015:**
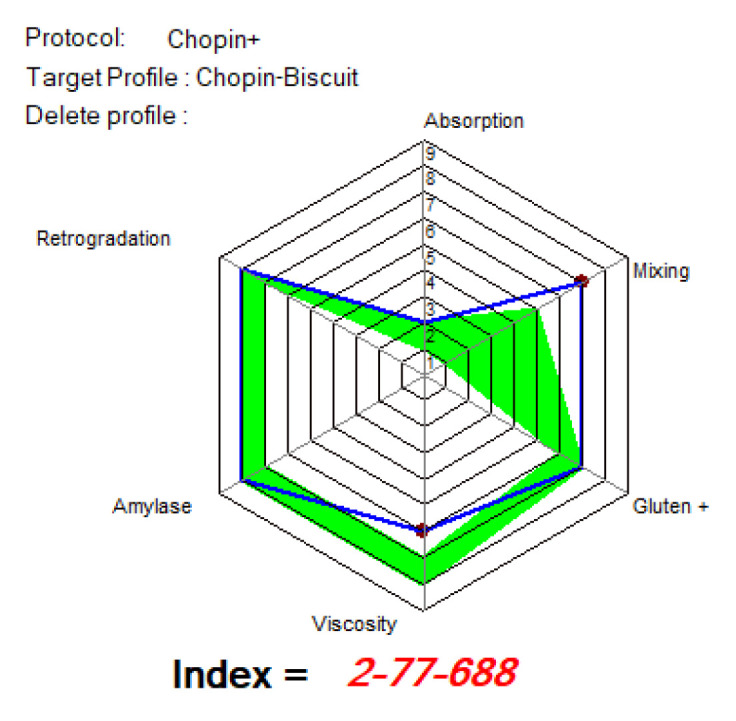
MIXOLAB Profiler index of the analyzed sample with 100% wheat flour (WF) for biscuits technology. Blue line represents the profile of composite flours and green line represents the profile of optimal MIXOLAB parameters for bread technology.

**Figure 16 foods-14-00319-f016:**
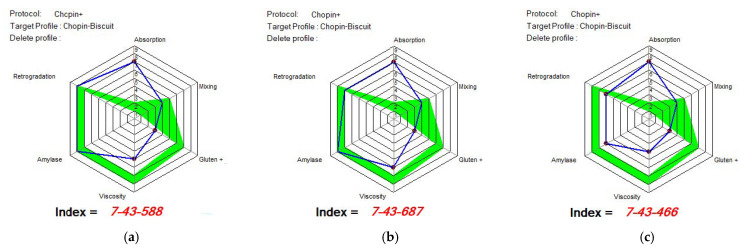
MIXOLAB Profiler index of the composite flours with different proportions of black lentil sprouts flour and wheat flour type 650 for biscuits technology. (**a**) BLWF 1, (**b**) BLWF2, (**c**) BLWF3. Blue line represents the profile of composite flours and green line represents the profile of optimal MIXOLAB parameters for bread technology.

**Figure 17 foods-14-00319-f017:**
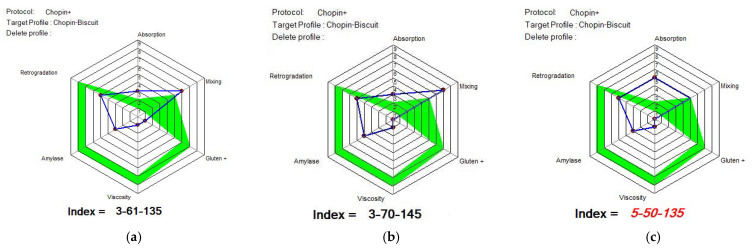
MIXOLAB Profiler index of the composite flours with different proportions of black lentil sprouts flour and wheat flour type 650 for biscuits technology. (**a**) SLWF 1, (**b**) SLWF2, (**c**) SLWF3. Blue line represents the profile of composite flours and green line represents the profile of optimal MIXOLAB parameters for bread technology.

**Figure 18 foods-14-00319-f018:**
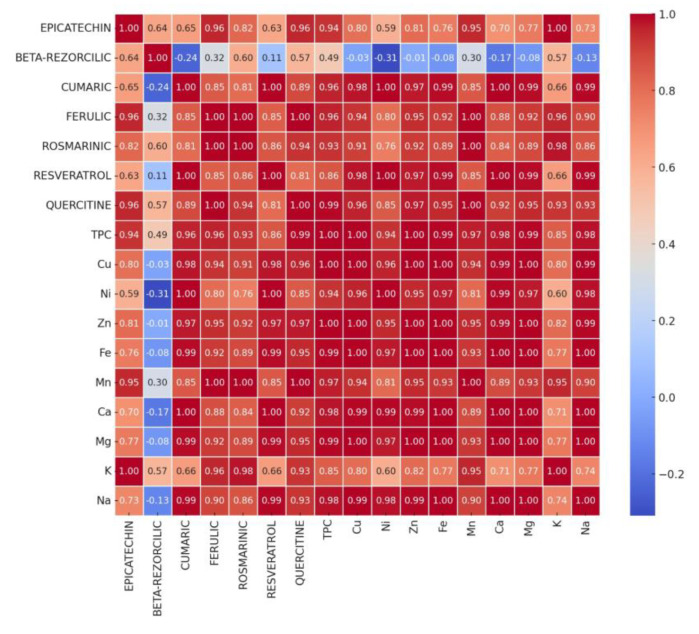
Pearson correlation between individual polyphenol contents and macro and microelement contents for composite flours BLWF1, BLWF2, and BLWF3.

**Figure 19 foods-14-00319-f019:**
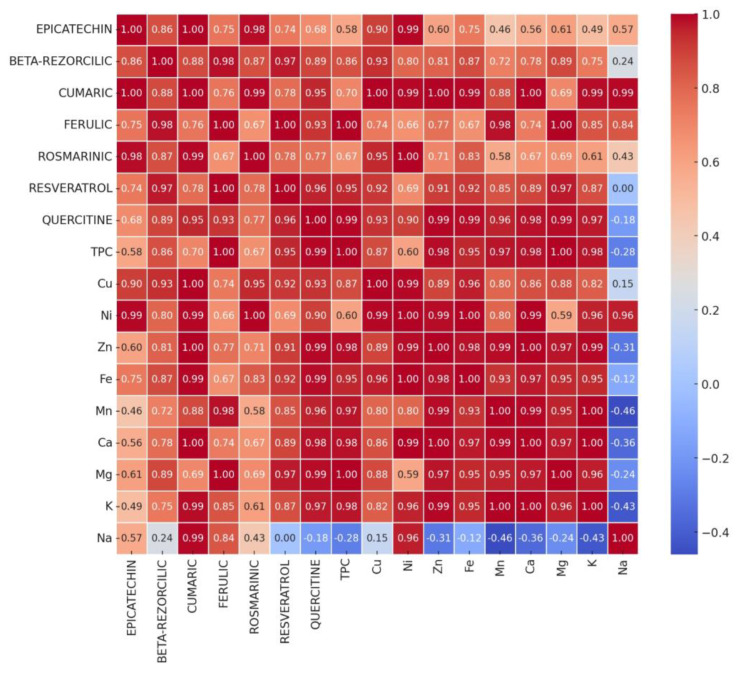
Pearson correlation between individual polyphenol contents and macro and microelement contents for composite flours SLWF1, SLWF2, and SLWF3.

**Figure 20 foods-14-00319-f020:**
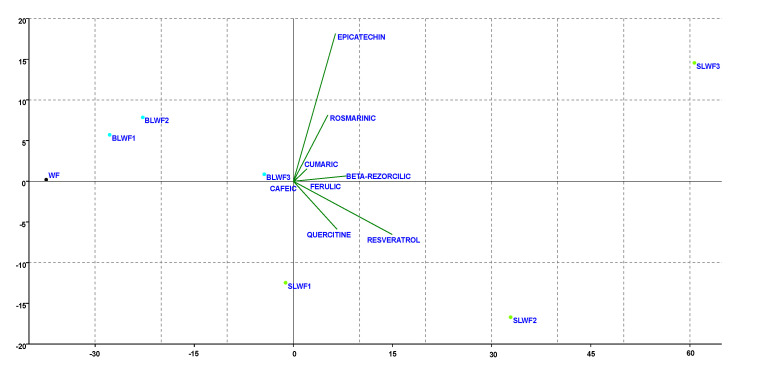
Projection of the parameters (individual polyphenols) of composite flours (BLWF and SLWF) by the first and second principal components.

**Figure 21 foods-14-00319-f021:**
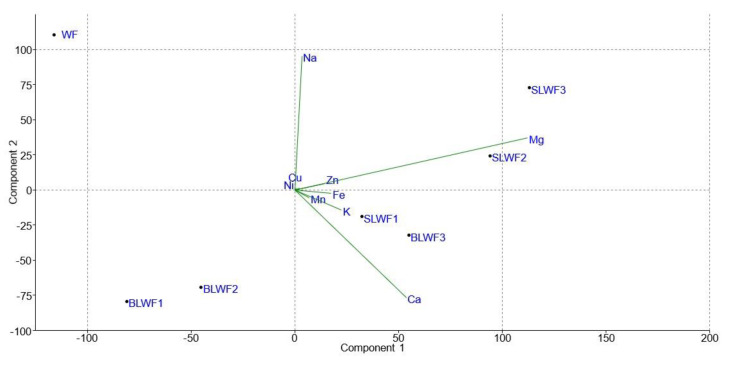
Projection of the parameters (macro and microelements) of composite flours (BLWF and SLWF) by the first and second principal components.

**Table 1 foods-14-00319-t001:** Proximate composition of composite flours.

Samples	Nutritional Characteristics				
	Moisture	Ash	Proteins	Lipids	Carbohydrates
	(%)	(%)	(%)	(%)	(g/100 g)
WF	11.43 ± 0.39 ^a^	0.5 ± 0.02 ^a^	11.14 ± 0.07 ^a^	1.30 ± 0.02 ^a^	75.63 ^a^
LF	10.00 ± 0.20 ^d^	3.76 ± 0.25 ^h^	25.00 ± 2.00 ^h^	0.84 ± 0.12 ^h^	60.39 ^i^
SLF	9.74 ± 0.71 ^d^	5.80 ± 0.1 ^d^	19.90 ± 0.5 ^g^	1.17 ± 0.10 ^e^	63.40 ^h^
**Lentil/Wheat composite flours**					
BLWF1	10.77 ± 0.06 ^b^	1.88 ± 0.09 ^e^	12.86 ± 0.02 ^d^	0.97 ± 0.02 ^e^	73.53 ^e^
BLWF2	10.36 ± 0.15 ^c^	2.18 ± 0.04 ^f^	13.98 ± 0.09 ^e^	0.77 ± 0.03 ^f^	72.72 ^f^
BLWF3	9.92 ± 0.13 ^d^	2.62 ± 0.03 ^g^	15.06 ± 0.13 ^f^	0.74 ± 0.02 ^g^	71.66 ^g^
**Sprouted lentil/Wheat composite flours**					
SLWF1	11.7 ± 0.01 ^a^	2.12 ± 0.04 ^b^	11.70 ± 0.11 ^a^	1.043 ± 0.002 ^b^	73.43 ^b^
SLWF2	11.39 ± 0.04 ^a^	3.82 ± 0.04 ^c^	12.34 ± 0.01 ^b^	1.05 ± 0.002 ^c^	71.40 ^c^
SLWF3	11.86 ± 0.02 ^a^	5.51 ± 0.16 ^d^	13.31 ± 0.17 ^c^	1.07 ± 0.002 ^d^	68.24 ^d^

WF-wheat flour, LF-lentil flour, SLF-lentil sprouts flour, BLWF1–3-composite wheat–lentil flours, SLWF1–3-composite wheat–lentil sprouts flours. The values are expressed as mean values ± standard deviations of all measurements; data within the same column sharing different superscripts are significantly different (*p* < 0.05); data within the same column sharing the same superscripts are not significantly different (*p* > 0.05).

**Table 2 foods-14-00319-t002:** Individual polyphenols of composite flours (mg/kg).

SAMPLE	Galic	Epicatechin	Cafeic	Beta-Rezorcilic	Ccumaric	Ferulic	Rosmarinic	Resveratrol	Quercitin
WF	nd *	5.53 ± 0.02 ^a^	nd *	3.77 ± 0.02 ^a^	nd *	nd *	1.386 ± 0.02 ^a^	2.371 ± 0.01 ^a^	2.399 ± 0.001 ^a^
BLWF1	nd *	15.179 ± 0.03 ^b^	nd *	3.88 ± 0.03 ^b^	3.489 ± 0.04 ^a^	2.316 ± 0.1 ^a^	2.614 ± 0.01 ^b^	7.274 ± 0.2 ^b^	7.744 ± 0.06 ^b^
BLWF2	nd *	17.769 ± 0.02 ^c^	nd *	4.615 ± 0.4 ^c^	3.536 ± 0.03 ^b^	3.32 ± 0.01 ^b^	6.53 ± 0.01 ^c^	9.623 ± 0.3 ^c^	10.456 ± 0.02 ^c^
BLWF3	nd *	18.112 ± 0.01 ^d^	nd *	4.037 ± 0.01 ^d^	4.012 ± 0.12 ^c^	3.981 ± 0.02 ^c^	8.49 ± 0.04 ^d^	34.086 ± 0.8 ^d^	12.869 ± 0.12 ^d^
SLWF1	nd *	6.78 ± 0.02 ^e^	nd *	10.57 ± 0.10 ^e^	3.490 ± 0.02 ^d^	3.78 ± 0.02 ^d^	7.558 ± 0.04 ^e^	28.576 ± 0.007 ^e^	31.08 ± 0.003 ^e^
SLWF2	nd *	13.612 ± 0.01 ^f^	nd *	26.025 ± 0.12 ^f^	3.721 ± 1.2 ^e^	19.204 ± 0.01 ^e^	8.444 ± 0.2 ^f^	57.063 ± 0.02 ^f^	38.03 ± 0.01 ^f^
SLWF3	nd *	46.979 ± 0.02 ^g^	nd *	35.28 ± 0.1 ^g^	4.789 ± 0.2 ^f^	22.65 ± 0.02 ^f^	30.059 ± 0.02 ^g^	64.693 ± 0.004 ^g^	45.956 ± 0.02 ^g^

WF-wheat flour, BLWF1–3-composite wheat–lentil flours, SLWF1–3-composite wheat–lentil sprouts flours. The values are expressed as mean values ± standard deviations of all measurements; data within the same column sharing different superscripts are significantly different (*p* < 0.05); data within the same column sharing the same superscripts are not significantly different (*p* > 0.05). * nd—not detectable.

**Table 3 foods-14-00319-t003:** The macro and microelements of composite flours.

Samples	Macro- and Microelement Contents (mg/kg)								
	Cu	Ni	Zn	Fe	Mn	Ca	Mg	K	Na
Composite flours									
WF	2.831 ± 0.9 ^a^	nd *	14.399 ± 0.9 ^a^	11.855 ± 1.2 ^a^	2.088 ± 0.3 ^a^	142.091 ± 0.2 ^a^	401.211 ± 1.3 ^a^	504.647 ± 2.8 ^a^	190.096 ± 0.9 ^a^
BLWF1	3.125 ± 0.01 ^e^	0.10 ± 0.014 ^d^	19.380 ± 0.16 ^e^	27.684 ± 0.02 ^e^	14.050 ± 0.00 ^e^	302.367 ± 1.13 ^e^	419.525 ± 1.13 ^e^	553.837 ± 1.1 ^e^	69.131 ± 0.007 ^e^
BLWF2	3.704 ± 0.00 ^f^	0.114 ± 0.034 ^e^	22.720 ± 0.01 ^f^	32.857 ± 0.09 ^f^	17.390 ± 0.04 ^f^	305.872 ± 0.21 ^f^	453.428 ± 0.94 ^f^	554.194 ± 1.84 ^f^	75.671 ± 0.007 ^f^
BLWF3	5.106 ± 0.01 ^g^	0.929 ± 0.002 ^f^	29.980 ± 0.01 ^g^	48.597 ± 0.01 ^g^	19.700 ± 0.01 ^g^	324.777 ± 0.03 ^g^	554.27 ± 1.85 ^g^	556.993 ± 2.4 ^g^	102.28 ± 1.253 ^g^
SLWF1	5.260 ± 0.00 ^b^	1.920 ± 0.002 ^a^	39.940 ± 0.05 ^b^	40.800 ± 0.06 ^b^	18.660 ± 1.74 ^b^	281.884 ± 0.02 ^b^	549.578 ± 1.15 ^b^	555.477 ± 2.0 ^b^	86.655 ± 0.01 ^b^
SLWF2	5.770 ± 0.01 ^c^	1.931 ± 0.004 ^b^	41.190 ± 0.01 ^c^	41.570 ± 0.03 ^c^	19.030 ± 0.02 ^c^	285.238 ± 0.58 ^c^	637.333 ± 4.08 ^c^	556.806 ± 2.2 ^c^	121.798 ± 0.04 ^c^
SLWF3	8.570 ± 0.00 ^d^	2.248 ± 0.007 ^c^	46.350 ± 0.01 ^d^	56.390 ± 0.08 ^d^	19.250 ± 0.04 ^d^	305.635 ± 0.15 ^d^	623.333 ± 2.88 ^d^	559.627 ± 2.5 ^d^	201.813 ± 0.06 ^d^

WF-wheat flour, BLWF1–3-composite wheat–lentil flours, SLWF1–3-composite wheat–lentil sprouts flours. The values are expressed as mean values ± standard deviations of all measurements; data within the same column sharing different superscripts are significantly different (*p* < 0.05); data within the same column sharing the same superscripts are not significantly different (*p* > 0.05). * nd—not detectable.

## Data Availability

The The original data presented in the study are openly available at the University of Life Sciences “King Mihai I” from Timișoara.

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
