# Peer review of "Composite Flours Based on Black Lentil Seeds and Sprouts with Nutritional, Phytochemical and Rheological Impact on Bakery/Pastry Products"

_foods, 2025, doi:10.3390/foods14020319_

Round 1

Reviewer 1 Report

Comments and Suggestions for Authors

Abstract

The abstract describes the study objective, the primary methods, and the important findings. Authors should include specific data showing the results obtained, such as in lines 29-33.

Introduction.

Although the introduction describes some generalities about lentils, fermentation, sprouting, bread, and products, the authors are advised to explain in more detail the work that preceded the present research to clarify the knowledge gap and the novelty of the present study.

Methodology.

Authors are advised to clarify the experimental design. Acronyms are recommended, but some letters referring to sprouted flour should be in the same sense; Figure 2 should be reorganized and included. Figure 3 seems repetitive.

In nutritional content studies, authors are advised to include the characterization of the lentils and sprouted flours used before mixing them.

Line 193: The authors explain using sodium carbonate "to neutralize the reaction." There may be confusion here. The authors are advised to review this detail.

Results

Table 1: The authors are advised to explain the acronyms used in column 1. Just so you know, the authors are advised to include the results for the lentil and sprout flours before mixing with the wheat flour.

Tables 2 and 3 are shown as figures and are not in a table format. The authors are advised to address this issue. The authors are advised to review the word "Quercitine" in Table 2.

The discussion of each identified phytochemical is extensive. The authors are advised to review this section.

The authors are advised to identify the colored lines in the graphs in Figures 7, 8, and 9.

Some discussion sections are extensive and seem only descriptive or comparative with other authors. Authors are encouraged to explain and highlight the importance of their findings.

Author Response

Dear Reviewer,

We would like to address all our thanks and gratitude for the constructive observations, corrections and recommendations.

Based on the reviewers’ recommendations, the authors of this paper responded point by point to the following aspects:

Comments and Suggestions for Authors

Comment 1: Abstract. The abstract describes the study objective, the primary methods, and the important findings. Authors should include specific data showing the results obtained, such as in lines 29-33.

Answer 1: In the abstract have been inserted specific data showing the obtained results

 Comment 2: Introduction. Although the introduction describes some generalities about lentils, fermentation, sprouting, bread, and products, the authors are advised to explain in more detail the work that preceded the present research to clarify the knowledge gap and the novelty of the present study.

Answer 2: The innovative and distinct character of the present study compared to other previous works was underlined at the end of the introduction.

 Comment 3: Methodology. Authors are advised to clarify the experimental design. Acronyms are recommended, but some letters referring to sprouted flour should be in the same sense; Figure 2 should be reorganized and included. Figure 3 seems repetitive.

Answer 3: The abbreviations were carefully checked in order to keep the same notation throughout the paper. Figures 2-3 have been merged.

 Comment 4: In nutritional content studies, authors are advised to include the characterization of the lentils and sprouted flours used before mixing them.

Answer 4: The nutritional values of lentil flours and sprouts before mixing were added in the table 1

 Comment 5: Line 193: The authors explain using sodium carbonate "to neutralize the reaction." There may be confusion here. The authors are advised to review this detail.

Answer 5: The correction was done.

 Comment 6: Results.Table 1: The authors are advised to explain the acronyms used in column 1. Just so you know, the authors are advised to include the results for the lentil and sprout flours before mixing with the wheat flour.

Answer 6. The explanation of abbreviation used in table 1 was added. The results for lentil and sprouts flours were included in the table 1.

Comment 7. Tables 2 and 3 are shown as figures and are not in a table format. The authors are advised to address this issue. The authors are advised to review the word "Quercitine" in Table 2.

Answer 7: The corrections were done

Comment 8: The discussion of each identified phytochemical is extensive. The authors are advised to review this section.

Answer 8: The discussion section was reviewed.

 Comment 9: The authors are advised to identify the colored lines in the graphs in Figures 7, 8, and 9.

Answer 9: The explanations of colored lines in the MIXOLAB figures were added

Comment 10: Some discussion sections are extensive and seem only descriptive or comparative with other authors. Authors are encouraged to explain and highlight the importance of their findings.

Answer 10: The discussions were adapted according to the recommendations made

 Once again, we would like to thank the reviewer for your appreciations, corrections and recommendations which contributed to the significant improvement of the paper.

Reviewer 2 Report

Comments and Suggestions for Authors

1.      Use of Punctuation. Punctuation is not used or is used incorrectly in several places in the article. For example, L58, L85, L282, etc.

2.      Use of grammar and units. For example, in L191, “5 minute” should be “5 minutes”, and in L209, “320” should be “320nm”

3.      L261. Table 1 shows that the moisture content of SLWF2 is lower than WF, not “lower than the samples with sprouted black lentil flour,” as the article states.

4.      L262. According to the data in Table 1 of the article, the moisture content of SLWF should be “11.39~11.86”.

5.      L305. The percentage decrease in carbohydrates for SLWF shown in Figure 4 is “2.5 to 9.4%”.

6.      L210 states “The results were expressed in mg/g”, but in the analysis of the results they were expressed as “mg/kg”.

7.      L330. The total polyphenol content of SWLF3 depicted in Figure 5 should be “791.237 mg/100g”.

8.      Are L365~L367 referenced, if so please give specific references.

9.      There are several instances in the article where the data descriptions do not match the charts. For example, the highest Ca content in lentil germ flour in L485 is “324.777”. For example, L490, L492.

10.   The maximum addition of black lentil flour or sprouted black lentil flour to the fortified wheat samples mentioned in the article was 20%, while fortified wheat flour with a 30% addition appeared several times in the article, e.g., L488, L498, L502, and many other places.

Author Response

Dear Reviewer,

We would like to address all our thanks and gratitude for the constructive observations, corrections and recommendations.

Based on the reviewers’ recommendations, the authors of this paper responded point by point to the following aspects:

 Comments and Suggestions for Authors

 Comment 1.      Use of Punctuation. Punctuation is not used or is used incorrectly in several places in the article. For example, L58, L85, L282, etc.

Answer 1: The punctuation has been checked throughout the manuscript

Comment 2.      Use of grammar and units. For example, in L191, “5 minute” should be “5 minutes”, and in L209, “320” should be “320nm”

Answer 2: The corrections were done

Comment 3.      L261. Table 1 shows that the moisture content of SLWF2 is lower than WF, not “lower than the samples with sprouted black lentil flour,” as the article states.

Answer 3: The correction was done

Comment 4.      L262. According to the data in Table 1 of the article, the moisture content of SLWF should be “11.39~11.86”.

Answer 4: The correction was done

Comment 5.      L305. The percentage decrease in carbohydrates for SLWF shown in Figure 4 is “2.5 to 9.4%”.

Answer 5: Thank you for the observation. The correction was done

Comment 6.      L210 states “The results were expressed in mg/g”, but in the analysis of the results they were expressed as “mg/kg”.

Answer 6: Thank you for the observation. The correction was done

Comment 7:     L330. The total polyphenol content of SWLF3 depicted in Figure 5 should be “791.237 mg/100g”.

Answer 7: Thank you for the observation. The correction was done

Comment 8.      Are L365~L367 referenced, if so please give specific references.

Answer 8: We apologize, it was a mistake, the results referred to the values obtained for wheat flour (WF), it was corrected in the text

Comment 9.      There are several instances in the article where the data descriptions do not match the charts. For example, the highest Ca content in lentil germ flour in L485 is “324.777”. For example, L490, L492.

Answer 9: Thank you for the observations. The corrections were done

Comment 10.   The maximum addition of black lentil flour or sprouted black lentil flour to the fortified wheat samples mentioned in the article was 20%, while fortified wheat flour with a 30% addition appeared several times in the article, e.g., L488, L498, L502, and many other places.

Answer 10: We apologize, you are right, the maximum substitution was 20%, this error was corrected everywhere in the text.

 Once again, we would like to thank the reviewer for your appreciations, corrections and recommendations which contributed to the significant improvement of the paper.

Round 2

Reviewer 1 Report

Comments and Suggestions for Authors

Although the authors made minor revisions, some details remain in the manuscript.

The discussion of results remains extensive. For example, discussing the bioactive properties of each identified polyphenol makes the discussion extensive and may be considered unnecessary information. It may be considered important for a review article. In the present manuscript, such bioactive properties were not determined. The authors are advised to review this section again and consider the relevance of such information.

Author Response

Comments and Suggestions for Authors

Although the authors made minor revisions, some details remain in the manuscript.

The discussion of results remains extensive. For example, discussing the bioactive properties of each identified polyphenol makes the discussion extensive and may be considered unnecessary information. It may be considered important for a review article. In the present manuscript, such bioactive properties were not determined. The authors are advised to review this section again and consider the relevance of such information.

Answer

The authors have reformulated the results and discussions section as required, in particular section 3.2. concerning bioactive compounds and 3.3. regarding the content of macro and microelements.

 Once again, we would like to thank the reviewer for your appreciations, corrections and recommendations which contributed to the significant improvement of the paper.